# CONSERVATIVE BAYESIAN MODEL-BASED VALUE EXPANSION FOR OFFLINE POLICY OPTIMIZATION

**Jihwan Jeong**[1,3]*, **Xiaoyu Wang**[1]*, **Michael Gimelfarb**[1,3], **Hyunwoo Kim**[2]†, **Baher Abdulhai**[1] **& Scott Sanner**[1,3]†

[1]University of Toronto, [2]LG AI Research, [3]Vector Institute
`{jihwan.jeong,cnxiaoyu.wang,mike.gimelfarb}@mail.utoronto.ca,`
`hwkim@lgresearch.ai, baher.abdulhai@utoronto.ca,`
`ssanner@mie.utoronto.ca`

## ABSTRACT

Offline reinforcement learning (RL) addresses the problem of learning a performant policy from a fixed batch of data collected by following some behavior policy. Model-based approaches are particularly appealing in the offline setting since they can extract more learning signals from the logged dataset by learning a model of the environment. However, the performance of existing model-based approaches falls short of model-free counterparts, due to the compounding of estimation errors in the learned model. Driven by this observation, we argue that it is critical for a model-based method to understand when to trust the model and when to rely on model-free estimates, and how to act conservatively w.r.t. both. To this end, we derive an elegant and simple methodology called conservative Bayesian model-based value expansion for offline policy optimization (CBOP), that trades off model-free and model-based estimates during the policy evaluation step according to their epistemic uncertainties, and facilitates conservatism by taking a lower bound on the Bayesian posterior value estimate. On the standard D4RL continuous control tasks, we find that our method significantly outperforms previous model-based approaches: e.g., MOPO by 116.4%, MOReL by 23.2% and COMBO by 23.7%. Further, CBOP achieves state-of-the-art performance on 11 out of 18 benchmark datasets while doing on par on the remaining datasets.

## 1 INTRODUCTION

Fueled by recent advances in supervised and unsupervised learning, there has been a great surge of interest in data-driven approaches to reinforcement learning (RL), known as *offline RL* (Levine et al., 2020). In offline RL, an RL agent must learn a good policy entirely from a logged dataset of past interactions, without access to the real environment. This paradigm of learning is particularly useful in applications where it is prohibited or too costly to conduct online trial-and-error explorations (e.g., due to safety concerns), such as autonomous driving (Yu et al., 2018), robotics (Kalashnikov et al., 2018), and operations research (Boute et al., 2022).

However, because of the absence of online interactions with the environment that give correcting signals to the learner, direct applications of *online* off-policy algorithms have been shown to fail in the *offline* setting (Fujimoto et al., 2019; Kumar et al., 2019; Wu et al., 2019; Kumar et al., 2020). This is mainly ascribed to the distribution shift between the learned policy and the *behavior policy* (data-logging policy) during training. For example, in $Q$-learning based algorithms, the distribution shift in the policy can incur uncontrolled overestimation bias in the learned value function. Specifically, positive biases in the $Q$ function for out-of-distribution (OOD) actions can be picked up during policy maximization, which leads to further deviation of the learned policy from the behavior policy, resulting in a vicious cycle of value overestimation. Hence, the design of offline RL algorithms revolves around how to counter the adverse impacts of the distribution shift while achieving improvements over the data-logging policy.

---

*Equal contribution
†Corresponding authors

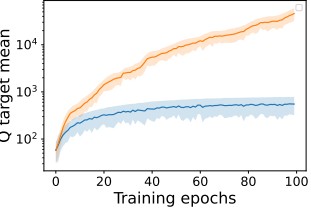 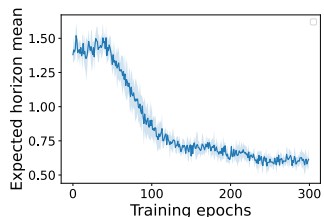

Figure 1: Prevention of value overestimation & adaptive reliance on model-based value predictions. (*Left*) We leverage the full posterior over the target values to prevent value overestimation during offline policy learning (blue). Without conservatism incorporated, the target value diverges (orange). (*Right*) We can automatically adjust the level of reliance on the model-based and bootstrapped model-free value predictions based on their respective uncertainty during model-based value expansion. The '*expected horizon*' ($\mathbb{E}[h] = \sum_h w_h \cdot h$, $\sum_h w_h = 1$) shows an effective model-based rollout horizon during policy optimization. $\mathbb{E}[h]$ is large at the beginning, but it gradually decreases as the model-free value estimates improve over time. The figures were generated using the *hopper-random* dataset from D4RL (Fu et al., 2020).

In this work, we consider model-based (MB) approaches since they allow better use of a given dataset and can provide better generalization capability (Yu et al., 2020; Kidambi et al., 2020; Yu et al., 2021; Argenson & Dulac-Arnold, 2021). Typically, MB algorithms — e.g., MOPO (Yu et al., 2020), MOReL (Kidambi et al., 2020), and COMBO (Yu et al., 2021) — adopt the Dyna-style policy optimization approach developed in online RL (Janner et al., 2019; Sutton, 1990). That is, they use the learned dynamics model to generate rollouts, which are then combined with the real dataset for policy optimization.

We hypothesize that we can make better use of the learned model by employing it for target value estimation during the policy evaluation step of the actor-critic method. Specifically, we can compute $h$-step TD targets through dynamics model rollouts and bootstrapped terminal $Q$ function values. In online RL, this MB value expansion (MVE) has been shown to provide a better value estimation of a given state (Feinberg et al., 2018). However, the naïve application of MVE does not work in the offline setting due to model bias that can be exploited during policy learning.

Therefore, it is critical to trust the model only when it can reliably predict the future, which can be captured by the epistemic uncertainty surrounding the model predictions. To this end, we propose CBOP (**C**onservative **B**ayesian MVE for **O**ffline **P**olicy Optimization) to control the reliance on the model-based and model-free value estimates according to their respective uncertainties, while mitigating the overestimation errors in the learned values. Unlike existing MVE approaches (e.g., Buckman et al. (2018)), CBOP estimates the *full* posterior distribution over a target value from the $h$-step TD targets for $h = 0, \ldots, H$ sampled from ensembles of the state dynamics and the $Q$ function. The novelty of CBOP lies in its ability to fully leverage this uncertainty in two related ways: (1) by deriving an adaptive weighting over different $h$-step targets informed by the posterior uncertainty; and (2) by using this weighting to derive *conservative* lower confidence bounds (LCB) on the target values that mitigates value overestimation. Ultimately, this allows CBOP to reap the benefits of MVE while significantly reducing value overestimation in the offline setting (Figure 1).

We evaluate CBOP on the D4RL benchmark of continuous control tasks (Fu et al., 2020). The experiments show that using the conservative target value estimate significantly outperforms previous model-based approaches: e.g., MOPO by 116.4%, MOReL by 23.2% and COMBO by 23.7%. Further, CBOP achieves state-of-the-art performance on 11 out of 18 benchmark datasets while doing on par on the remaining datasets.

## 2 BACKGROUND

We study RL in the framework of *Markov decision processes* (MDPs) that are characterized by a tuple $(\mathcal{S}, \mathcal{A}, T, r, d_0, \gamma)$; here, $\mathcal{S}$ is the state space, $\mathcal{A}$ is the action space, $T(\mathbf{s}'|\mathbf{s}, \mathbf{a})$ is the transition function, $r(\mathbf{s}, \mathbf{a})$ is the immediate reward function, $d_0$ is the initial state distribution, and $\gamma \in [0, 1]$ is the discount factor. Specifically, we call the transition and reward functions the *model* of the environment, which we denote as $f = (T, r)$. A *policy* $\pi$ is a mapping from $\mathcal{S}$

to $\mathcal{A}$, and the goal of RL is to find an optimal policy $\pi^*$ which maximizes the expected cumulative discounted reward, $\mathbb{E}_{\mathbf{s}_t, \mathbf{a}_t}[\sum_{t=0}^{\infty} \gamma^t r(\mathbf{s}_t, \mathbf{a}_t)]$, where $\mathbf{s}_0 \sim d_0, \mathbf{s}_t \sim T(\cdot|\mathbf{s}_{t-1}, \mathbf{a}_{t-1})$, and $\mathbf{a}_t \sim \pi^*(\cdot|\mathbf{s}_t)$. Often, we summarize the quality of a policy $\pi$ by the state-action value function $Q^\pi(\mathbf{s}, \mathbf{a}) := \mathbb{E}_{\mathbf{s}_t, \mathbf{a}_t}[\sum_{t=0}^{\infty} \gamma^t r(\mathbf{s}_t, \mathbf{a}_t)|\mathbf{s}_0 = \mathbf{s}, \mathbf{a}_0 = \mathbf{a}]$, where $\mathbf{a}_t \sim \pi(\cdot|\mathbf{s}_t) \, \forall t > 0$.

Off-policy actor-critic methods, such as SAC (Haarnoja et al., 2018) and TD3 (Fujimoto et al., 2018), have enjoyed great successes in complex continuous control tasks in deep RL, where parameterized neural networks for the policy $\pi_\theta$ (known as actor) and the action value function $Q_\phi$ (known as critic) are maintained. Following the framework of the *generalized policy iteration* (GPI) (Sutton & Barto, 2018), we understand the actor-critic algorithm as iterating between (i) policy evaluation and (ii) policy improvement. Here, policy evaluation typically refers to the calculation of $Q_\phi(\mathbf{s}, \pi_\theta(\mathbf{s}))$ for the policy $\pi_\theta$, while the improvement step is often as simple as maximizing the currently evaluated $Q_\phi$; i.e., $\max_\theta \mathbb{E}_{\mathbf{s} \sim \mathcal{D}}[Q_\phi(\mathbf{s}, \pi_\theta(\mathbf{s}))]$ (Fujimoto et al., 2018).

**Policy Evaluation**  At each iteration of policy learning, we evaluate the current policy $\pi_\theta$ by minimizing the *mean squared Bellman error* (MSBE) with the dataset $\mathcal{D}$ of previous state transitions:

$$\mathcal{L}(\phi, \mathcal{D}) = \text{MSBE} := \mathbb{E}_{(\mathbf{s}, \mathbf{a}, r, \mathbf{s}') \sim \mathcal{D}}\left[ (y(\mathbf{s}, \mathbf{a}, \mathbf{s}') - Q_\phi(\mathbf{s}, \mathbf{a}))^2 \right], \tag{1}$$

$$y(\mathbf{s}, \mathbf{a}, \mathbf{s}') = r(\mathbf{s}, \mathbf{a}) + \gamma Q_{\phi'}(\mathbf{s}', \mathbf{a}'), \ \ \mathbf{a}' \sim \pi_\theta(\cdot|\mathbf{s}') \tag{2}$$

where $y(\mathbf{s}, \mathbf{a}, \mathbf{s}')$ is the *TD target* at each $(\mathbf{s}, \mathbf{a})$, towards which $Q_\phi$ is regressed. A separate *target network* $Q_{\phi'}$ is used in computing $y$ to stabilize learning (Mnih et al., 2015). Off-policy algorithms typically use some variations of (2), e.g., by introducing the clipped double-$Q$ trick (Fujimoto et al., 2018), in which $\min_{j=1,2} Q_{\phi'_j}(\mathbf{s}', \mathbf{a}')$ is used instead of $Q_{\phi'}(\mathbf{s}', \mathbf{a}')$ to prevent value overestimation.

**Model-based Offline RL**  In the offline setting, we are given a fixed set of transitions, $\mathcal{D}$, collected by some *behavior* policy $\pi_\beta$, and the aim is to learn a policy $\pi$ that is better than $\pi_\beta$. In particular, offline *model-based* (MB) approaches learn the model $\hat{f} = (\hat{T}, \hat{r})$ of the environment using $\mathcal{D}$ to facilitate the learning of a good policy. Typically, $\hat{f}$ is trained to maximize the log-likelihood of its predictions. Though MB algorithms are often considered capable of better generalization than their *model-free* (MF) counterparts by leveraging the learned model, it is risky to trust the model for OOD samples. Hence, MOPO (Yu et al., 2020) and MOReL (Kidambi et al., 2020) construct and learn from a pessimistic MDP where the model uncertainty in the next state prediction is penalized in the reward. Criticizing the difficulty of accurately computing well-calibrated model uncertainty, COMBO (Yu et al., 2021) extends CQL (Kumar et al., 2020) to the model-based regime by regularizing the value function on OOD samples generated via model rollouts. These methods follow the *Dyna-style* policy learning where model rollouts are used to augment the offline dataset (Sutton, 1990; Janner et al., 2019).

**Model-based Value Expansion (MVE) for Policy Optimization**  An alternative to the aforementioned *Dyna-style* approaches is MVE (Feinberg et al., 2018), which is arguably better suited to seamlessly integrating the power of both MF and MB worlds. In a nutshell, MVE attempts to more accurately estimate the TD target in (2) by leveraging a model of the environment, which can lead to more efficient policy iteration. Specifically, we can use the $h$-step MVE target $\hat{R}_h(\mathbf{s}, \mathbf{a}, \mathbf{s}')$ for $y(\mathbf{s}, \mathbf{a}, \mathbf{s}')$:

$$\hat{y}(\mathbf{s}, \mathbf{a}, \mathbf{s}') = \hat{R}_h(\mathbf{s}, \mathbf{a}, \mathbf{s}') := \sum_{t=0}^{h} \gamma^t \hat{r}_t(\hat{\mathbf{s}}_t, \hat{\mathbf{a}}_t) + \gamma^{h+1} Q_{\phi'}(\hat{\mathbf{s}}_{h+1}, \hat{\mathbf{a}}_{h+1}), \tag{3}$$

$$(\hat{\mathbf{s}}_0, \hat{\mathbf{a}}_0, \hat{r}_0, \hat{\mathbf{s}}_1) = (\mathbf{s}, \mathbf{a}, r, \mathbf{s}'), \ \hat{\mathbf{s}}_t \sim \hat{T}(\cdot|\hat{\mathbf{s}}_{t-1}, \hat{\mathbf{a}}_{t-1}), \ \hat{\mathbf{a}}_t \sim \pi_\theta(\cdot|\hat{\mathbf{s}}_t), \ 1 \le t \le h+1,$$

where $\hat{R}_h(\mathbf{s}, \mathbf{a}, \mathbf{s}')$ is obtained by the $h$-step MB return plus the terminal value at $h + 1$ ($h = 0$ reduces back to MF). In reality, errors in the learned model $\hat{f}$ compound if rolled out for a large $h$. Thus, it is standard to set $h$ to a small number.

## 3   CONSERVATIVE BAYESIAN MVE FOR OFFLINE POLICY OPTIMIZATION

The major limitations of MVE when applied to offline RL are as follows:

1. The model predictions $\hat{\mathbf{s}}_t$ and $\hat{r}_t$ in (3) become increasingly less accurate as $t$ increases because model errors can compound, leading to largely biased target values. This issue is exacerbated in the offline setup because we cannot obtain additional experiences to reduce the model error.

2. The most common sidestep to avoid the issue above is to use short-horizon rollouts only. However, rolling out the model for only a short horizon even when the model can be trusted could severely restrict the benefit of being model-based.

3. Finally, when the model rollouts go outside the support of $\mathcal{D}$, $\hat{R}_h$ in (3) can have a large overestimation bias, which will eventually be propagated into the learned $Q_\phi$ function.

Ideally, we want to **control the reliance on the model $\hat{f}$ and the bootstrapped $Q_{\phi'}$ according to their respective epistemic uncertainty, while also preventing $Q_\phi$ from accumulating large overestimation errors**. That is, when we can trust $\hat{f}$, we can safely roll out the model for more steps to get a better value estimation. On the contrary, if the model is uncertain about the future it predicts, we should shorten the rollout horizon and bootstrap from $Q_{\phi'}$ early on. Indeed, Figure 1 (right) exemplifies that CBOP relies much more on the MB rollouts at the beginning of training because the value function is just initialized. As $Q_{\phi'}$ becomes more accurate over time, CBOP automatically reduces the weights assigned to longer MB rollouts.

Below, we present CBOP, a Bayesian take on achieving the aforementioned two goals: trading off the MF and MB value estimates based on their uncertainty while obtaining a conservative estimation of the target $\hat{y}(\mathbf{s}, \mathbf{a}, \mathbf{s}')$. To this end, we first let $\hat{Q}^\pi(\mathbf{s}_t, \mathbf{a}_t)$ denote the value of the policy $\pi$ at $(\mathbf{s}_t, \mathbf{a}_t)$ in the learned MDP defined by its dynamics $\hat{f}$; that is,

$$\hat{Q}^\pi(\mathbf{s}_t, \mathbf{a}_t) = \mathbb{E}_{\hat{f}, \pi}\left[\sum_{k=0}^{\infty} \gamma^k \hat{r}(\hat{\mathbf{s}}_{t+k}, \hat{\mathbf{a}}_{t+k})\right], \ (\hat{\mathbf{s}}_t, \hat{\mathbf{a}}_t) = (\mathbf{s}_t, \mathbf{a}_t), \ \hat{\mathbf{a}}_{t+k} \sim \pi(\cdot|\hat{\mathbf{s}}_{t+k}). \quad (4)$$

Note that in the offline MBRL setting, we typically cannot learn $Q^\pi$ due to having only an approximation $\hat{f}$ of the model, and thus we focus instead on learning $\hat{Q}^\pi$.

Although there exists a unique $\hat{Q}^\pi(\mathbf{s}, \mathbf{a})$ at each $(\mathbf{s}, \mathbf{a})$ given a fixed model $\hat{f}$, we cannot directly observe the value unless we infinitely roll out the model from $(\mathbf{s}, \mathbf{a})$ until termination, which is computationally infeasible. Instead, we view each $\hat{R}_h \ \forall h$ defined in (3) as a conditionally independent

---

**Algorithm 1** Conservative Bayesian MVE

**Input:** $(\mathbf{s}_t, \mathbf{a}_t, r_t, \mathbf{s}_{t+1}), \hat{f}, Q_{\phi'}$
1. Sample $\hat{R}_h \ \forall h \leq H$ using $\hat{f}$ and $Q_{\phi'}$ as in (3)
2. Estimate $\mu_h, \ \sigma_h$ according to (8), (9)
3. Compute the posterior $\mathcal{N}(\mu, \sigma)$ using (7)
**return** conservative value target (e.g., LCB $\mu - \psi\sigma$)

---

(biased) noisy observation of the true underlying parameter $\hat{Q}^\pi$.[1] From this assumption, we can construct the Bayesian posterior over $\hat{Q}^\pi$ given the observations $\hat{R}_h \ \forall h$. With the closed-form posterior distribution at hand, we can take various conservative estimates from the distribution; we use the lower confidence bound (LCB) in this work. Algorithm 1 summarizes the procedure at a high-level. Please see Algorithm 2 in Appendix B.1 for the full description of CBOP.

### 3.1 CONSERVATIVE VALUE ESTIMATION VIA BAYESIAN INFERENCE

In this part, we formally discuss the conservative value estimation of CBOP based on Bayesian posterior inference. Specifically, the parameter of interest is $\hat{Q}^\pi$, and we seek its posterior estimation:

$$\mathbb{P}\left(\hat{Q}^\pi \mid \hat{R}_0, \ldots, \hat{R}_H\right) \propto \mathbb{P}\left(\hat{R}_0, \ldots, \hat{R}_H \mid \hat{Q}^\pi\right) \mathbb{P}\left(\hat{Q}^\pi\right) = \mathbb{P}\left(\hat{Q}^\pi\right) \prod_{h=0}^{H} \mathbb{P}\left(\hat{R}_h \mid \hat{Q}^\pi\right), \quad (5)$$

where we assume that $\hat{R}_h \ (h = 0, \ldots, H)$ are conditionally independent given $\hat{Q}^\pi$ (see Appendix A where we discuss in detail about the assumptions present in CBOP).

In this work, we model the likelihood of observations $\mathbb{P}(\hat{R}_h|\hat{Q}^\pi)$ as normally distributed with the mean $\mu_h$ and the standard deviation $\sigma_h$:

$$\hat{R}_h \mid \hat{Q}^\pi \sim \mathcal{N}(\mu_h, \sigma_h^2), \quad (6)$$

---

[1]We will omit $(\mathbf{s}, \mathbf{a}, \mathbf{s}')$ henceforth if it is clear from the context.

since it leads to a closed-form posterior update. Furthermore, since $\hat{R}_h$ can be seen as a sum of future immediate rewards, when the MDP is ergodic and $\gamma$ is close to 1, the Gaussian assumption (approximately) holds according to the central limit theorem (Dearden et al., 1998). Also, note that our Bayesian framework is not restricted to the Gaussian assumptions, and other surrogate probability distributions such as the Student-t distribution could be used instead.

For the prior, we use the improper (or uninformative) prior, $\mathbb{P}(\hat{Q}^\pi) = 1$, since it is natural to assume that we lack generally applicable prior information over the target value across different environments and tasks (Christensen et al., 2011). The use of the improper prior is well justified in the Bayesian literature (Wasserman, 2010; Berger, 1985), and the particular prior we use in CBOP corresponds to the Jeffreys prior, which has the invariant property under a change of coordinates. The Gaussian likelihood and the improper prior lead to a 'proper' Gaussian posterior density that integrates to 1, from which we can make various probabilistic inferences (Wasserman, 2010).

The posterior (5) is a Gaussian with mean $\mu$ and variance $\sigma^2$, defined as follows:

$$\rho = \sum_{h=0}^{H} \rho_h, \quad \mu = \sum_{h=0}^{H} \left( \frac{\rho_h}{\sum_{h=0}^{H} \rho_h} \right) \mu_h, \quad (7)$$

where $\rho = 1/\sigma^2$ and $\rho_h = 1/\sigma_h^2$ are the precisions of the posterior and the likelihood of $\hat{R}_h$, respectively. The posterior mean $\mu$ corresponds to the MAP estimation of $\hat{Q}^\pi$. Note that $\mu$ has the form of a weighted sum, $\sum_h w_h \mu_h$, with $w_h = \rho_h / \sum_{h=0}^{H} \rho_h \in (0, 1)$ being the weight allocated to $\hat{R}_h$. If the variance of $\hat{R}_h$ for some $h$ is relatively

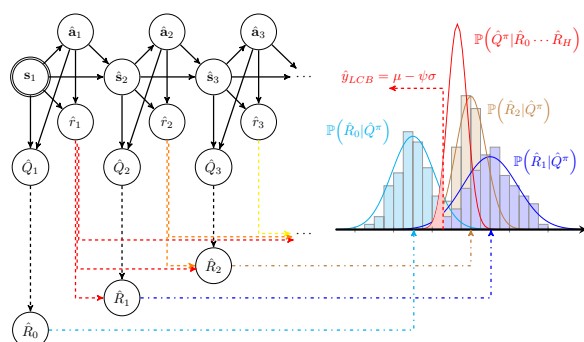

Figure 2: The graphical model representation of CBOP

large, we give a smaller weight to that observation. If, on the other hand, $\hat{R}_h$ all have the same variance (e.g. $\rho_0 = \cdots = \rho_H$), we recover the usual $H$-step return estimate. Recall that the quality of $\hat{R}_h$ is determined by that of the model rollout return and the bootstrapped terminal value. Thus intuitively speaking, the adaptive weight $w_h$ given by the Bayesian posterior allows the trade-off between the epistemic uncertainty of the model with that of the $Q$ function.

Figure 2 illustrates the overall posterior estimation procedure. Given a transition tuple $(\mathbf{s}, \mathbf{a}, r, \mathbf{s}')$, we start the model rollout from $\mathbf{s}_1 = \mathbf{s}'$. At each rollout horizon $h$, the cumulative discounted reward $\sum_{t=0}^{h} \gamma^t \hat{r}_t$ is sampled by the dynamics model and the terminal value $\hat{Q}_h$ is sampled by the $Q$ function (the sampling procedure is described in Section 3.2). We then get $\hat{R}_h$ by adding the $h$-step MB return samples and the terminal values $\gamma^{h+1} \hat{Q}_{h+1}$, which we deem as sampled from the distribution $\mathbb{P}(\hat{R}_h | \hat{Q}^\pi)$ parameterized by $\mu_h, \sigma_h^2$ (we use the sample mean and variance). These individual $h$-step *observations* are then combined through the Bayesian inference to give us the posterior distribution over $\hat{Q}^\pi$.

It is worth noting that the MAP estimator can also be derived from the perspective of variance optimization (Buckman et al., 2018) over the target values. However, we have provided much evidence in Section 4 and Appendix D.3 that the point estimate does not work in the offline setting due to value overestimation. Hence, it is imperative that we should have the full posterior distribution over the target value, such that we can make a conservative estimation rather than the MAP estimation.

To further understand the impact of using the MAP estimator for the value estimation, consider an estimator $\tilde{Q}$ of $\hat{Q}^\pi$ and its squared loss: $L(\hat{Q}^\pi, \tilde{Q}) = (\hat{Q}^\pi - \tilde{Q})^2$. It is known that the posterior mean of $\hat{Q}^\pi$ minimizes the *Bayes risk* w.r.t. $L(\hat{Q}^\pi, \tilde{Q})$ (Wasserman, 2010), meaning that the posterior risk $\int L(\hat{Q}^\pi, \tilde{Q}) \mathbb{P}(\hat{Q}^\pi | \hat{R}_0, \ldots, \hat{R}_H) d\hat{Q}^\pi$ is minimized at $\tilde{Q} = \mu$. In this context, $\mu$ is also called the (generalized) *Bayes estimator* of $\hat{Q}^\pi$, which is an admissible estimator (Robert, 2007). Despite seemingly advantageous, this result has a negative implication in offline RL. That is, the MAP estimator minimizes the squared loss from $\hat{Q}^\pi(\mathbf{s}, \mathbf{a})$ over the entire support of the posterior, weighted by the posterior distribution. Now, the distribution shift of $\pi$ from $\pi_\beta$ can lead to significantly biased $\hat{Q}^\pi$ compared to the true $Q^\pi$. In this case, the quality of the MAP estimator when

evaluated in the real MDP would be poor. Especially, the overestimation bias in the MAP estimation can quickly propagate to the $Q_\phi$ function and thereby exacerbate the distribution shift.

## 3.2 ENSEMBLES OF DYNAMICS AND Q FUNCTIONS FOR SAMPLING H-STEP MVE TARGETS

In this section, we discuss how we estimate the parameters $\mu_h, \sigma_h^2$ of $\mathbb{P}(\hat{R}_h|\hat{Q}^\pi)$ from the ensemble of dynamics models and that of $Q$ functions.

Assume we have a bootstrapped dynamics ensemble model $\hat{f}$ consisting of $K$ different models $(\hat{f}_1, \ldots, \hat{f}_K)$ trained with different sequences of mini-batches of $\mathcal{D}$ (Chua et al., 2018; Janner et al., 2019). Similarly, we assume a $Q$ ensemble of size $M$. Given a state $\hat{s}_t$ and an action $\hat{a}_t$, we can construct the probability over the next state $\hat{s}_{t+1}$ and reward $\hat{r}_t$ by the ensemble as follows:

$$\mathbb{P}(\hat{\mathbf{s}}_{t+1}, \hat{r}_t|\hat{\mathbf{s}}_t, \mathbf{a}_t) = \sum_{k=1}^{K} \mathbb{P}(\hat{f}_k) \cdot \mathbb{P}(\hat{\mathbf{s}}_{t+1}, \hat{r}_t|\hat{\mathbf{s}}_t, \mathbf{a}_t, \hat{f}_k)$$

where $\mathbb{P}(\hat{f}_k)$ is the probability of selecting the $k$th model from the ensemble, which is $1/K$ when all models are weighted equally. Now, the sampling method that exactly follows the probabilistic graphical model shown in Figure 2 would first sample a model from the ensemble at each time step, followed by sampling the next state transition (and reward) from the model, which should then be repeated $K$ times per state to generate a single sample. Then, we evaluate the resulting state $\hat{s}_{t+1}$ and action $\hat{a}_{t+1} \sim \pi_\theta(\hat{s}_{t+1})$ with the $Q$ ensemble to obtain $M$ samples. To obtain $N$ trajectories from a single initial state to estimate $\mu_h$ and $\sigma_h^2$ for $h = 1, \ldots, H$, the overall procedure requires $\mathcal{O}(NKH)$ computation, which can quickly become infeasible for moderately large $K$ and $N$ values.

To reduce the computational complexity, we follow Chua et al. (2018) where each *particle* is propagated by a single model of the ensemble for $H$ steps. With this, we can obtain $N$ trajectories of length $H$ from one state with $\mathcal{O}(NH)$ instead of $\mathcal{O}(NKH)$ (below we use $N = K$, i.e., we generate one particle per model). Concretely, given a single transition $\tau = (\mathbf{s}_0, \mathbf{a}_0, r_0, \mathbf{s}_1)$, we create $K$ numbers of *particles* by replicating $\mathbf{s}_1$ $K$ times, denoted as $\hat{\mathbf{s}}_1^{(k)} \forall k$. The $k$th particle is propagated by a fixed model $\hat{f}_k$ and the policy $\pi_\theta$ for $H$ steps, where $(\hat{\mathbf{s}}_t^{(k)}, \hat{r}_{t-1}^{(k)}) = \hat{f}_k(\hat{\mathbf{s}}_{t-1}^{(k)}, \hat{\mathbf{a}}_{t-1}^{(k)})$ and $\hat{\mathbf{a}}_t^{(k)} \sim \pi_\theta(\hat{\mathbf{s}}_t^{(k)})$. At each imagined timestep $t \in [0, H+1]$, $M$ number of terminal values are sampled by the $Q_{\phi'}$ ensemble at $(\hat{\mathbf{s}}_t^{(k)}, \hat{\mathbf{a}}_t^{(k)})$.

Despite the computational benefit, an implication of this sampling method is that it no longer directly follows the graphical model representation in Figure 2. However, we can still correctly estimate $\mu_h$ and $\sigma_h^2$ by turning to the law of total expectation and the law of total variance. That is,

$$\mu_h = \mathbb{E}_{\pi_\theta}\left[\hat{R}_h \mid \tau\right] = \mathbb{E}_{\hat{f}_k}\left[\mathbb{E}_{\pi_\theta}\left[\hat{R}_h \mid \tau, \hat{f}_k\right]\right] \tag{8}$$

where the outer expectation is w.r.t. the dynamics ensemble sampling probability $\mathbb{P}(\hat{f}_k) = 1/K$. Hence, given a fixed dynamics model $\hat{f}_k$, we sample $\hat{R}_h$ by following $\pi_\theta$ and compute the average of the $h$-step return, which is then averaged across different ensemble models. In fact, the resulting $\mu_h$ is the mean of all aggregated $M \times K$ samples of $\hat{R}_h$.

The $h$-step return variance $\mathrm{Var}_{\pi_\theta}(\hat{R}_h|\tau)$ decomposes via the law of total variance as following:

$$\sigma_h^2 = \mathrm{Var}_{\pi_\theta}\left[\hat{R}_h|\tau\right] = \underbrace{\mathbb{E}_{\hat{f}_k}\left[\mathrm{Var}_{\pi_\theta}\left[\hat{R}_h|\tau, \hat{f}_k\right]\right]}_{A} + \underbrace{\mathrm{Var}_{\hat{f}_k}\left[\mathbb{E}_{\pi_\theta}\left[\hat{R}_h \mid \tau, \hat{f}_k\right]\right]}_{B}. \tag{9}$$

Here, $A$ is related to the epistemic uncertainty of the $Q_{\phi'}$ ensemble; while $B$ is associated with the epistemic uncertainty of the dynamics ensemble. The total variance $\mathrm{Var}_{\pi_\theta}(\hat{R}_h|\tau)$ captures both uncertainties. This way, even though we use a different sampling scheme than presented in the graphical model of Figure 2, we can compute the unbiased estimators of the Gaussian parameters.

Once we obtain $\mu_h$ and $\sigma_h^2$, we plug them into (7) to compute the posterior mean and the variance. A conservative value estimation can be made by $\hat{y}_{LCB} = \mu - \psi\sigma$ with some coefficient $\psi > 0$ (Jin et al., 2021; Rashidinejad et al., 2021). Under the Gaussian assumption, this corresponds to the worst-case return estimate in a Bayesian credible interval for $\hat{Q}^\pi$. We summarize CBOP in Algorithm 2 in Appendix B.1.

Table 1: Normalized scores on D4RL MuJoCo Gym environments. Experiments ran with 5 seeds.

|  |  | MOPO | MOReL | COMBO | CQL | TD3+BC | EDAC | IQL | CBOP |
|---|---|---|---|---|---|---|---|---|---|
| random | halfcheetah | $35.4 \pm 2.5$ | 25.6 | **38.8** | 35.4 | $10.2 \pm 1.3$ | $28.4 \pm 1.0$ | - | $32.8 \pm 0.4$ |
| | hopper | $11.7 \pm 0.4$ | **53.6** | 17.9 | 10.8 | $11.0 \pm 0.1$ | $31.3 \pm 0.0$ | - | $31.4 \pm 0.0$ |
| | walker2d | $13.6 \pm 2.6$ | 37.3 | 7.0 | 7.0 | $1.4 \pm 1.6$ | $\mathbf{21.7 \pm 0.0}$ | - | $17.8 \pm 0.4$ |
| medium | halfcheetah | $42.3 \pm 1.6$ | 42.1 | 54.2 | 44.4 | $42.8 \pm 0.3$ | $67.5 \pm 1.2$ | 47.4 | $\mathbf{74.3 \pm 0.2}$ |
| | hopper | $28.0 \pm 12.4$ | 95.4 | 94.9 | 79.2 | $99.5 \pm 1.0$ | $101.6 \pm 0.6$ | 66.2 | $\mathbf{102.6 \pm 0.1}$ |
| | walker2d | $17.8 \pm 19.3$ | 77.8 | 75.5 | 58.0 | $79.7 \pm 1.8$ | $92.5 \pm 0.8$ | 78.3 | $\mathbf{95.5 \pm 0.4}$ |
| medium replay | halfcheetah | $53.1 \pm 2.0$ | 40.2 | 55.1 | 46.2 | $43.3 \pm 0.5$ | $63.9 \pm 0.8$ | 44.2 | $\mathbf{66.4 \pm 0.3}$ |
| | hopper | $67.5 \pm 24.7$ | 93.6 | 73.1 | 48.6 | $31.4 \pm 3.0$ | $101.8 \pm 0.5$ | 94.7 | $\mathbf{104.3 \pm 0.4}$ |
| | walker2d | $39.0 \pm 9.6$ | 49.8 | 56.0 | 26.7 | $25.2 \pm 5.1$ | $87.1 \pm 2.3$ | 73.8 | $\mathbf{92.7 \pm 0.9}$ |
| medium expert | halfcheetah | $63.3 \pm 38.0$ | 53.3 | 90.0 | 62.4 | $97.9 \pm 4.4$ | $\mathbf{107.1 \pm 2.0}$ | 86.7 | $105.4 \pm 1.6$ |
| | hopper | $23.7 \pm 6.0$ | 108.7 | 111.1 | 98.7 | $\mathbf{112.2 \pm 0.2}$ | $110.7 \pm 0.1$ | 91.5 | $111.6 \pm 0.2$ |
| | walker2d | $44.6 \pm 12.9$ | 95.6 | 96.1 | 111.0 | $101.1 \pm 9.3$ | $114.7 \pm 0.9$ | 109.6 | $\mathbf{117.2 \pm 0.5}$ |
| expert | halfcheetah | - | - | - | - | $105.7 \pm 1.9$ | $\mathbf{106.8 \pm 3.4}$ | - | $100.4 \pm 0.9$ |
| | hopper | - | - | - | - | $\mathbf{112.2 \pm 0.2}$ | $110.3 \pm 0.3$ | - | $111.4 \pm 0.2$ |
| | walker2d | - | - | - | - | $105.7 \pm 2.7$ | $115.1 \pm 1.9$ | - | $\mathbf{122.7 \pm 0.8}$ |
| full replay | halfcheetah | - | - | - | - | - | $84.6 \pm 0.9$ | - | $\mathbf{85.5 \pm 0.3}$ |
| | hopper | - | - | - | - | - | $105.4 \pm 0.7$ | - | $\mathbf{108.1 \pm 0.3}$ |
| | walker2d | - | - | - | - | - | $99.8 \pm 0.7$ | - | $\mathbf{107.8 \pm 0.2}$ |

## 4 EXPERIMENTS

We have designed the experiments to answer the following research questions: **(RQ1)** Is CBOP able to adaptively determine the weights assigned to different $h$-step returns according to the relative uncertainty of the learned model and that of the $Q$ function? **(RQ2)** How does CBOP perform in the offline RL benchmark? **(RQ3)** Does CBOP with LCB provide conservative target $Q$ estimation? **(RQ4)** How does having the full posterior over the target values compare against using the MAP estimation in performance? **(RQ5)** How much better is it to adaptively control the weights to $h$-step returns during training as opposed to using a fixed set of weights throughout training?

We evaluate these RQs on the standard *D4RL* offline RL benchmark (Fu et al., 2020). In particular, we use the D4RL MuJoCo Gym dataset that contains three environments: *halfcheetah*, *hopper*, and *walker2d*. For each environment, there are six different behavior policy configurations: *random* (*r*), *medium* (*m*), *medium-replay* (*mr*), *medium-expert* (*me*), *expert* (*e*), and *full-replay* (*fr*). We release our code at `https://github.com/jihwan-jeong/CBOP`.

### 4.1 CBOP CAN AUTOMATICALLY ADJUST RELIANCE ON THE LEARNED MODEL

To investigate RQ1, we use the notion of the *expected rollout horizon*, which we define as $\mathbb{E}[h] = \sum_{h=0}^{H} w_h \cdot h$. Here, $w_h$ is the weight given to the mean of $\hat{R}_h$ as defined in (7), which sums to 1. A larger $\mathbb{E}[h]$ indicates that more weights are assigned to longer-horizon model-based rollouts.

Figure 1 already shows that $\mathbb{E}[h]$ decreases as the $Q$ function becomes better over time. On the other hand, Figure 3 shows how the quality of the learned model affects $\mathbb{E}[h]$. Specifically, we trained the dynamics model on *halfcheetah-m* for different numbers of epochs $(10, \ldots, 100)$; then, we trained the policy with CBOP for 150 epochs.

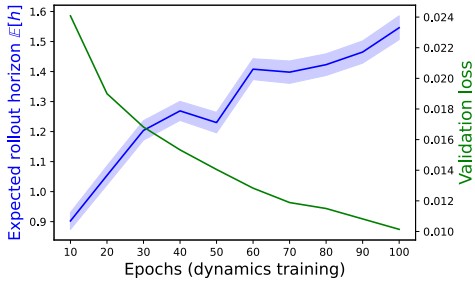

Figure 3: $\mathbb{E}[h]$ during CBOP training with the dynamics model trained for different numbers of epochs. CBOP can place larger weights to longer-horizon rollouts as the dynamics model becomes more accurate.

### 4.2 PERFORMANCE COMPARISON

To investigate RQ2, we select baselines covering both model-based and model-free approaches: (*model-free*) **CQL** (Kumar et al., 2020), **IQL** (Kostrikov et al., 2022), **TD3+BC** (Fujimoto & Gu, 2021), **EDAC** (An et al., 2021); (*model-based*) **MOPO** (Yu et al., 2020), **MOReL** (Kidambi et al., 2020), and **COMBO** (Yu et al., 2021). Details of experiments are provided in Appendix C.1.

Table 1 shows the experimental results. Comparing across all baselines, CBOP presents new state-of-the-art performance in **11 tasks out of 18** while performing similar in the remaining configurations. Notably, CBOP outperforms prior works in *medium*, *medium-replay*, and *full-replay* configurations with large margins. We maintain that these are the datasets of greater interest than, e.g., *random* or *expert* datasets because the learned policy needs to be much different than the behavior policy in order to perform well. Furthermore, the improvement compared to previous model-based arts is substantial: CBOP outperforms MOPO, MOReL, and COMBO by 116.4%, 23.2% and 23.7% (respectively) on average across four behavior policy configurations.

## 4.3 CBOP Learns Conservative Values

To answer RQ3, we have selected 3 configurations (*m*, *me*, and *mr*) from the *hopper* environment and evaluated the value function at the states randomly sampled from the datasets, i.e., $\mathbb{E}_{\mathbf{s} \sim \mathcal{D}}[\hat{V}^{\pi}(\mathbf{s})]$ (nb. a similar analysis is given in CQL). Then, we compared these estimates with the Monte Carlo estimations from the true environment by rolling out the learned policy until termination.

Table 2: Difference between the values predicted by the learned $Q$ functions and the true discounted returns from the environment.

| Task name | CQL | | CBOP | |
|---|---|---|---|---|
| | Mean | Max | Mean | Max |
| hopper-m | -61.84 | -3.20 | -55.83 | -16.21 |
| hopper-mr | -142.89 | -28.73 | -172.45 | -39.45 |
| hopper-me | -79.67 | -5.16 | -114.39 | -11.24 |

Table 2 reports how large are the value predictions compared to the true returns. Notice that not only the mean predictions are negative but also the maximum values are, which affirms that CBOP indeed has learned conservative value functions. Despite the predictions by CBOP being smaller than those of CQL in *hopper-mr* and *me*, we can see that CBOP significantly outperforms CQL in these settings. See Appendix D.1 for more details.

## 4.4 Ablation Studies

**LCB vs. MAP in the offline setting**  To answer RQ4, we compare CBOP with STEVE (Buckman et al., 2018) which is equivalent to using the MAP estimation for target $Q$ predictions. Figure 1 (left) shows the case where the value function learned by STEVE blows up (orange). Further, we include the performance of STEVE in all configurations in Appendix D.3. To summarize the results, STEVE fails to learn useful policies for 11 out of 18 tasks. Especially, except for the *fr* datasets, using the MAP estimation has led to considerable drops in the performances in the *hopper* and *walker2d* environments, which reaffirms that it is critical to have the full posterior distribution over the target values such that we can make conservative target predictions.

**Adaptive weighting**  For RQ5, we also considered an alternative way of combining $\hat{R}_h \; \forall h$ by explicitly assigning a fixed set of weights: uniform or geometric. We call the latter $\lambda$-weighting, in reference to the idea of TD($\lambda$) (Sutton, 1988). We evaluated the performance of the fixed weighting scheme with various $\lambda \in (0, 1)$ values, and report the full results in Appendix D.3. In summary, there are some $\lambda$ values that work well in a specific task. However, it is hard to pick a single $\lambda$ that works across all environments, and thus $\lambda$ should be tuned as a hyperparameter. In contrast, CBOP can avoid this problem by automatically adapting the rollout horizon.

**Benefits of full posterior estimation**  To ablate the benefits of using the full posterior distribution in conservative policy optimization, we have compared CBOP to a quantile-based approach that calculates the conservative estimate through the $\alpha$-quantile of the sampled returns $\hat{y}(\mathbf{s}, \mathbf{a}, \mathbf{s}')$ (3) from the ensemble. The experimental details and results are reported in Appendix D.3. In summary, we have found that CBOP consistently outperformed this baseline on all tasks considered, and CBOP was more stable during training, showing the effectiveness of the Bayesian formulation.

## 5 Related Work

In the pure offline RL setting, it is known that the direct application of off-policy algorithms fails due to value overestimation and the resulting policy distribution shift (Kumar et al., 2019; 2020; Fujimoto & Gu, 2021; Yu et al., 2021). Hence, it is critical to strike the balance between *conservatism* and *generalization* such that we mitigate the extent of policy distribution shift while ensuring

that the learned policy $\pi_\theta$ performs better than behavior policy $\pi_\beta$. Below, we discuss how existing model-free and model-based methods address these problems in practice.

**Model-free offline RL** *Policy constraint* methods directly constrain the deviation of the learned policy from the behavior policy. For example, BRAC (Wu et al., 2019) and BEAR (Kumar et al., 2019) regularize the policy by minimizing some divergence measure between these policies (e.g., MMD or KL divergence). Alternatively, BCQ (Fujimoto et al., 2019) learns a generative model of the behavior policy and uses it to sample perturbed actions during policy optimization. On the other hand, *value regularization* methods such as CQL (Kumar et al., 2020) add regularization terms to the value loss in order to implicitly regulate the distribution shift (Kostrikov et al., 2021; Wang et al., 2020). Recently, some simple yet effective methods have been proposed. For example, TD3+BC (Fujimoto & Gu, 2021) adds a behavioral cloning regularization term to the policy objective of an off-policy algorithm (TD3) (Fujimoto et al., 2018) and achieves SOTA performances across a variety of tasks. Also, by extending Clipped Double Q-learning (Fujimoto et al., 2018) to an ensemble of $N$ $Q$ functions, EDAC (An et al., 2021) achieves good benchmark performances.

**Model-based offline RL** Arguably, the learning paradigm of offline RL strongly advocates the use of a dynamics model, trained in a supervised way with a fixed offline dataset. Although a learned model can help generalize to unseen states or new tasks, model bias poses a significant challenge. Hence, it is critical to know when to trust the model and when not to. MOPO (Yu et al., 2020) and MOReL (Kidambi et al., 2020) address this issue by constructing and learning from a pessimistic MDP whose reward is penalized by the uncertainty of the state prediction. On the other hand, COMBO (Yu et al., 2021) extends CQL within the model-based regime by regularizing the value function on OOD samples generated via model rollouts. Rigter et al. (2022) also takes an adversarial approach by optimizing the policy with respect to a worst-case dynamics model. In contrast to these, CBOP estimates a full Bayesian posterior over values by using ensembles of models and value functions during policy evaluation of an actor-critic algorithm. In principle, having the full distribution that CBOP provides could also facilitate the use of other risk-informed statistics and epistemic risk measures to address value overestimation (see, e.g., Eriksson & Dimitrakakis (2020)).

**Model-based value expansion** Unlike Dyna-style methods that augment the dataset with model-generated rollouts (Sutton, 1990; Janner et al., 2019), MVE (Feinberg et al., 2018) uses them for better estimating TD targets during policy evaluation. While equally weighted $h$-step model returns were used in MVE, STEVE (Buckman et al., 2018) introduced an adaptive weighting scheme from the optimization perspective by approximately minimizing the variance of the MSBE loss, while ignoring the bias. Interestingly, the Bayesian posterior mean (i.e., the MAP estimator) we derive in (7) matches the weighting scheme proposed in STEVE. However as we show in Figure 1 and 10, using the MAP estimator as value prediction in the offline setting often results in largely overestimated $Q$ values, which immensely hampers policy learning. See Section 3.1 for the related discussion.

# 6 CONCLUSION

In this paper, we present CBOP: conservative Bayesian model-based value expansion (MVE) for offline policy optimization. CBOP is a model-based offline RL algorithm that trades off model-free and model-based value estimates according to their respective epistemic uncertainty during policy evaluation while facilitating conservatism by taking a lower bound on the Bayesian posterior value estimate. Viewing each $h$-step MVE target as a conditionally independent noisy observation of the *true* target value under the learned MDP, we derive the Bayesian posterior distribution over the target value. For a practical implementation of CBOP, we use the ensemble of dynamics and that of $Q$ function to sample MVE targets to estimate the Gaussian parameters, which in turn are used to compute the posterior distribution. Through empirical and analytical analysis, we find that the MAP estimator of the posterior distribution could easily lead to value overestimation when the learned MDP is not accurate under the current policy. In contrast, CBOP constructs the LCB from the Bayesian posterior as a conservative estimation of the target value to successfully mitigate the issue while achieving state-of-the-art performance on several benchmark datasets.

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
