# OpenReview forum: "Conservative Bayesian Model-Based Value Expansion for Offline Policy Optimization"
_ICLR.cc/2023/Conference — ICLR 2023 poster_

### Official Review · Reviewer_yFv4 · 2022-10-25

**Confidence:** 3
**Correctness:** 3
**Technical Novelty And Significance:** 3
**Empirical Novelty And Significance:** 3
**Recommendation:** 6

**Clarity, Quality, Novelty And Reproducibility:**

The paper is generally clearly written, though the notation and discussion of the probabilistic model is hard to follow and not well supported. The experimental results however suggest the performance is impressive w.r.t. baselines and ablations, and the experiments are discussed clearly. From my perspective, the approach is original.

**Strength And Weaknesses:**

Strengths:
- The paper was well written, and explained concepts clearly. The background section provided the right level of information to the reader, explaining the key steps of generalized policy iteration, and how learned models can be used in the policy expansion phase.
- Leveraging the Bayesian posterior over Q values to construct a lower confidence bound for conservative exploration is an elegant way to incorporate conservatism and prevent overestimation bias as the state/action distribution deviates from that of the behavior policy.
- The experimental evaluation was thorough and demonstrated a significant performance improvement relative to baselines. I appreciated the comparisons to alternative weighting techniques, highlighting the benefit of the Bayesian inference based approach over simpler algorithmic choices like $\lambda$-weighting

Weaknesses:
- The discussion of the probabilistic model and assumptions was imprecise:
    - Section 3.1 seeks to define a posterior belief over $\hat{Q}^\pi$, the Q function for acting under $\hat{f}$ with policy $\pi$ and accruing rewards according to $\hat{r}$. However, then in section 3.2, this posterior is interpreted as a credible interval for $Q^\pi$, the Q function for the true dynamics, and then used to construct the targets for Q function regression. As the true Q function is true quantity of interest, it would make more sense to frame section 3.1 as estimating p(Q^pi \mid \hat{R}_0, …, \hat{R}_H), and likewise to treat $\hat{R}_h | Q^\pi$ as normally distributed, as opposed to $\hat{R}_h | \hat{Q}^\pi$.
    - The discussion motivating using the conservative estimate as opposed to the MAP estimator of the posterior belief in section 2 was hard to follow. Specifically, the last paragraph on page 5 felt unnecessary, as pointing out that $\mu$ is an admissible estimator of $\hat{Q}^\pi$ does not provide any evidence that a lower-confidence bound estimate would lead to a less biased estimate of the true Q function than the MAP estimate under the posterior.
    - The state-transition model using the ensemble dynamics model in section 3.2 conflates aleatoric and epistemic uncertainty by proposing that at each timestep, we first sample a dynamics model, and then sample a transition from the chosen model. This could lead to sample trajectories that are not actually dynamically feasible under any one of the individual models. The authors mention that in practice, they hold the model fixed over the course of the rollout when sampling for the purposes of reducing computational complexity. While this does reduce the sampling requirements, it also alters the probabilistic model to one that correctly separates uncertainty over a stationary dynamics model from stochasticity in individual transitions, as discussed in [Chua et al, 2018]. However, as the authors acknowledge in the Appendix, this breaks the conditional independence assumption they leverage to compute the Bayesian posterior.
    - It is unclear why the mean and variance computed according to (9) and (10) should yield the mean and variance of the observation model p( R_h | \hat{Q} ). Ensembles of neural networks can be viewed as performing approximate Bayesian inference, i.e. modeling p( f | D), where the prior p(f) depends on the initialization of the ensemble weights []. The authors should clarify why the predictive distribution when marginalizing over the two ensembles reflects the conditional distribution p( R_h | \hat{Q} ).
- Experiments were limited to deterministic environments:
    - While the derivation of the algorithm allowed for environments with stochastic transitions, the experiments focused on domains with deterministic transition and reward functions. I would be interested to see how stochasticity inherent to the environment would impact the performance of CBOP.
- Hyperparameters were tuned based on final online evaluation performance.
    - The reported results relied on different LCB coefficients for different environments, which as the authors report in the appendix C.2., was chosen based on final online evaluation performance. Given that this sort of hyperparameter optimization would not be possible in true safety-critical settings which motivate offline RL, it would have been useful to see the impact of this hyperparameter on performance. Specifically, I would imagine that as $\psi$ is increased from 0, we would see an trend that interpolates between the MAP based performance of STEVE  to the reported results in CBOP.
    - The initialization of the Q ensemble seemed to play a large part in CBOP's performance -- indeed, as the experiments demonstrated, using a randomly initialized Q ensemble led to poor results as the predictive variance was not calibrated. Furthermore, the authors leveraged gradient diversification to improve the performance of the algorithm. I would have liked to have seen a greater discussion about the importance of a calibrating variance estimates form the Q ensemble, i.e. ensuring the ensemble variance is an accurate estimate of $Var(R_0 | \hat{Q}^\pi))$ -- especially in the body of the text.

Minor comments:
- There are some notation mistakes / incomplete sentences, especially in the appendix. For example, on page 16, the definition of $Q_\phi^k = \arg\min_\phi …$ is not a valid equation, and the first paragraph of C.2 ends with a fragment.

**Summary Of The Paper:**

This paper proposes a method for model-based offline RL which leverages a learned model to perform model-based value expansion (MVE) in the policy evaluation phase of an actor critic style algorithm. Motivated by the risks of compounding errors and instability when using a model learned only from limited data from a behavior policy in offline RL, the authors propose a Bayesian approach which aims to automatically determine the degree to leverage the learned model, while also leveraging the posterior to apply conservatism to mitigate overestimation bias in the learned Q function. They demonstrate the approach outperforms model-based approaches on standard benchmarks, and achieves state-of-the-art performance on several domains.

**Summary Of The Review:**

Overall, I think the approach presents an interesting idea for combining the benefits of model-free and model-based methods for offline RL, and demonstrates that this idea can lead to impressive performance, which supports it's acceptance. However, the paper could be significantly improved with a clearer discussion of the probabilistic model and the assumptions made to translate the predictive distributions obtained through the DNN ensembles to the assumed measurement model on the true Q function.

---

> ### Author Response · Authors · 2022-11-19
> **Response to Reviewer yFv4 (1/2)**
>
> ### "This posterior is interpreted as a credible interval for $Q^\pi$"
>
> This is a typo in our draft and should be $\hat{Q}^\pi$. Thank you for pointing it out, and we have revised our manuscript appropriately.
>
>
> ### "As the true $Q$ function is the true quantity of interest"
>
>
> We want to clarify that we do not have access to the true dynamics of the MDP, but only a fixed set of data from which we can learn an approximate model. Therefore, since we can not estimate the true $Q^\pi$, the best we can do is to estimate $\hat{Q}^\pi$, the value corresponding to the learned dynamics model. Thank you for pointing this out, which we have clarified in our revision.
>
>
> ### "Does not provide any evidence that a lower-confidence bound estimate would lead to a less biased estimate"
>
>
> We reiterate one of the key challenges of policy optimization in the offline RL setting, namely that the exploitation of a poorly estimated model of the state dynamics can lead to overly confident or pathological policies. In other words, the MAP estimator is only good if the model dynamics are correct, which is too strong an assumption to hold in the offline regime. Therefore, prior work and our paper have argued that some degree of conservatism is necessary during policy learning. CBOP handles this in a novel way, by adaptive weighing the different-length rollouts from the model, such that the model is effectively ignored in favor of a bootstrap whenever the predictions of the model cannot be trusted (and vice versa). While we cannot control the bias in the learned model due to the limited offline data, we can lessen the reliance on a poorly estimated model by learning to avoid it if the Bayesian uncertainty is high. CBOP contributes two novel yet complementary ideas: (1) by weighting the rollout horizon adaptively using knowledge of the full posterior distribution, and (2) by taking a conservative estimation of the value function. Our experiments have empirically validated the necessity of both aspects to mitigate value overestimation in offline RL.
>
>
> ### "This could lead to sample trajectories that are not actually dynamically feasible"
>
>
> This is related to the point above. The novelty of CBOP is to avoid the use of the model if the predictions strongly are believed to lie significantly outside the training data distribution. By expressing this belief in a posterior distribution, we can then utilize it in the confidence bound to avoid training the policy on sampled trajectories that are not dynamically feasible. In addition, we do not believe the ensemble approach exacerbates the issue described because the risk of sampling dynamically infeasible trajectories can also happen in the single model case.  That is, unless a specific set of constraints are placed on the class of the dynamics model, a model learned via supervised learning with offline data cannot guarantee that it will always generate a dynamically-feasible trajectory.
>
>
> ### “However, as the authors acknowledge in the Appendix, this breaks the conditional independence assumption”
>
>
> We agree that one of the advantages of the particle-based approach is the separation of epistemic and aleatoric uncertainties of the model, as argued in Chua et al. (2018). However, we want to clarify that the goal of CBOP was focused on how to better estimate and utilize the epistemic uncertainty for value estimation in the offline RL setting, which is quite different from the aforementioned work. Our benchmarks (D4RL), which are quite standard in the offline setting and consist of deterministic dynamics, are also a reflection of this. We also want to reiterate that we found no benefit in enforcing a conditional independence assumption of the samples, although this could be easily implemented at the cost of increased computation.
>
>
> ### "Ensembles of neural networks can be viewed as performing approximate Bayesian inference, i.e. modeling p( f | D), where the prior p(f) depends on the initialization of the ensemble weights"
>
>
> We want to clarify that the goal of our paper is not to learn a weighting over ensemble predictions $p(f | D)$. Instead, the goal is to measure the confidence of the Bayesian ensemble by measuring the disagreement between the different model predictions $f_1, f_2,\dots, f_K$, which are trained separately on different batches of data sampled from the offline data set. Intuitively, if the different estimates of the $f_k$ agree (disagree) on the same input, then there is low (high) epistemic uncertainty in the predictions of the ensemble. It is common in the literature on Bayesian estimation to use a uniform sampling $p(f) = 1/K$, and this is what we have used in the paper. We agree, however, that our approach could also accommodate non-uniform prediction.
>
> (continued...)

---

> ### Author Response · Authors · 2022-11-19
> **Response to Reviewer yFv4 (2/2)**
>
> ### "The experiments focused on domains with deterministic transition"
>
> We agree that our approach could also accommodate learning stochastic transition models as well. However, we designed the current benchmarks to be consistent with those reported in the existing offline RL literature, in order to facilitate a consistent and fair evaluation with the state-of-the-art.
>
>
> ### “I would have liked to have seen a greater discussion about the importance of calibrating variance estimates”
>
>
> Thank you for pointing this out. Indeed, we agree that having well-calibrated estimates of value uncertainty is important, and this is also a key driver in many other aspects of RL such as exploration in the online RL setting. This is very important in the offline setting as well, as CBOP has shown, since value and model uncertainty are both critical for conservative estimation. Furthermore, the offline setting presents unique challenges such as dependence on initialization that must be taken into account. We argue that further developments in Bayesian learning from data will help CBOP and other works that also leverage the full Bayesian posterior for better value estimation.
>
>
> ### "$Q_\phi^k=argmin_\phi$... ends with a fragment"
>
> Thank you for pointing these out. They have been fixed in our revision.
>
>
> ### "Notation and discussion of the probabilistic model is hard to follow and not well supported"
>
>
> We hope that our responses to your questions above have eliminated some of the confusion about the ensemble estimation and sampling procedure used by CBOP. We will revisit the notation in the paper to address this. If you have any suggestions about how we can further improve the clarity of our exposition, please let us know during the discussion.

---

### Official Review · Reviewer_fgeG · 2022-10-25

**Confidence:** 4
**Correctness:** 3
**Technical Novelty And Significance:** 3
**Empirical Novelty And Significance:** 3
**Recommendation:** 8

**Clarity, Quality, Novelty And Reproducibility:**

The paper is well-written and the proposed idea is easy to follow. The authors have provided some details of their implementation, which, along with the source code provided in supplementary materials, allows for reproducing the results for further validation.

Questions:

- It seems that CBOP can be formulated as an extension to MOPO, and if so, its performance might suffer from the same reasons that affect the performance of MOPO. To be more specific, both MOPO and CBOP rely on the uncertainty quantified via an ensemble of networks to perform, so if by any chance, the networks within the ensemble are correlated to one another (which is not uncommon in practice), both methods may fail to perform. In the case of CBOP, correlation within an ensemble leads to an incorrect posterior estimation, which can hurt its performance in a similar way that it does in MOPO. I would appreciate a discussion on this matter, and what makes CBOP perform well in situations where MOPO fails for the reason mentioned above.

- From what I understand, as the training progresses and the value networks start to converge, the first term in equation (10) will start to fade. Hence, this term (which corresponds to the uncertainty among the Q-ensemble) will only affect the intermediary stages of the training, and not the final result itself. It would be interesting to see an ablation study where you investigate the effect of this term (which translates to the effect of utilizing a Q-ensemble instead of a Q-function) on the stability and convergence of CBOP.

- From the algorithm presented in the appendix, CBOP only optimizes the policy over the visited states in the offline dataset. Why did you not extend the policy optimization to out-of-distribution datapoints as well?

- Rigter et al. [1] present a model-based offline RL approach that does not depend on ensemble learning for uncertainty estimation. Could you clarify the similarities/differences/benefits between your approach and theirs?

[1] Rigter, M., Lacerda, B., & Hawes, N. (2022). Rambo-rl: Robust adversarial model-based offline reinforcement learning. arXiv preprint arXiv:2204.12581.


**Strength And Weaknesses:**

Strengths:
+ The paper is well written and clear to understand. I honestly enjoyed reading the paper.
+ The experiments show strong and promising results for CBOP compared to state-of-the-art.
+ The proposed method is novel (to some extent), straightforward, and easy to follow.

Weaknesses:
- The proposed method has not been theoretically analyzed. In my opinion, any theoretical analysis and guarantees would add great value to the current work.
- There are some parts in the paper that deserve to be discussed in more detail. For further elaboration, please refer to the questions below.
- Along with an ensemble of dynamics models, CBOP also utilizes an ensemble of value functions, which makes this method computationally more demanding than other state-of-the-art methods.
- Similar to MOPO, CBOP heavily relies on the diversity among the ensemble networks. As a result, if the networks within the ensemble turn out to be correlated to one another (which is not uncommon in practice), this method may potentially fail to work properly.

**Summary Of The Paper:**

The paper presents CBOP, a new approach for model-based offline reinforcement learning, where the RL agent is trained to optimize a lower bound on the Bayesian posterior estimate of values. In this approach, each h-step return from a particular state in the offline dataset is treated as a sample from the Bayesian posterior estimate of the value in that state, from which the posterior distribution over the target values is derived. The authors further balance model-free and model-based estimates of the values during policy evaluation by incorporating a lower bound on the Bayesian posterior value estimates for conservativity.



**Summary Of The Review:**

The paper appears to be solid, well-written and clear to understand. It proposes a novel method for model-based offline RL, which is straightforward and easy to follow. CBOP shows strong promise in the experiments compared to other SOTA methods. Although there are some comments that still need to be addressed to improve the paper, I vote for accepting the paper.

---

> ### Author Response · Authors · 2022-11-19
> **[1/2] Response to Reviewer fgeG**
>
> ### “Theoretical analysis and guarantees would add great value to the current work.”
>
> We thank the reviewer for this suggestion. We want to point out that the justification for LCB and the inspiration behind the adaptive weighting originated from prior theoretical developments (as discussed at the end of Section 3.2). We believe that one way to approach this in our setting could be through a robust MDP perspective (see, e.g. Iyengar 2005, Nilim and El Ghaoui 2005). Intuitively, we argue that CBOP can be roughly interpreted as maximizing a lower bound of the return under the assumption that the worst-case dynamics are selected from elliptical uncertainty sets (such as a Bayesian credible region). However, we will also need some strong assumptions on the learning algorithms used for policy evaluation and optimization to ensure convergence (e.g. linear). We do confirm experimentally that the Gaussian assumption for modeling return samples is in fact a good approximation in the benchmarks studied, but we believe generalizing and proving this formally will be much more challenging. If you have any suggestions on how we could proceed without making very restrictive assumptions, please let us know.
>
> ### "..., which makes this method computationally more demanding than other state-of-the-art methods."
>
> Indeed, you are correct that using ensembles increases the computation of CBOP. However, we contend that the benefit of having the epistemic uncertainty estimations outweighs the increased computation since it allows CBOP to leverage model and value uncertainty to make better-informed decisions about which of these estimates to trust. Also, all these computations are done purely offline, and we have shown experimentally that the increased computation translates to consistently better out-of-distribution performance of CBOP on testing than the state-of-the-art.
>
> ### "CBOP heavily relies on the diversity among the ensemble networks."
>
> MOPO estimates the uncertainty in the next state predictions directly and uses them as a penalty in the reward function. As discussed in Lu et al. (2022), this introduces various design choices as to which uncertainty estimation heuristics to use. CBOP, however, samples trajectories for rollouts of different lengths and adaptively weighs them to modulate the uncertainties in the model-based rollouts and the model-free value bootstraps. This adaptive approach makes it unnecessary to directly measure the uncertainty in the next state predictions.
>
> ### "If the networks within the ensemble turn out to be correlated."
>
> Please note that the approach for training the ensembles in our work closely follows previous work on Bayesian ensemble estimation (Chua et al., 2018, Janner et al., 2019). To reduce the effect of correlation, we follow the existing work by training each ensemble member using random batches from the dataset.
>
> We agree that sampling-based methods can suffer from correlation and that addressing this is a challenging and important research question at present. However, there are also key advantages to estimating uncertainty in this way. Firstly, bootstrapped uncertainty estimates have been shown to have strong theoretical properties (e.g., Efron 1982). Secondly, similar bootstrapping approaches have been shown to produce reasonably well-calibrated estimates of uncertainty in a number of distinct applications where deep function approximation is required (in both online RL such as bootstrapped DQN, and offline RL as cited above), without suffering from the potential computational intractability of other approaches in large state/action spaces. Our experiments also confirm that the obtained estimates obtained were well-calibrated in our setting. For further details, please see the expected horizon analysis (Figure 3 and Section 4.1), which demonstrates the effectiveness of CBOP subject to different qualities of the learned dynamics ensemble. We have also updated Appendix B.2 to reflect this discussion.
>
> __References__
>
> Garud N. Iyengar. Robust dynamic programming. Mathematics of Operations Research 30.2 (2005): 257-280.
>
> Arnab Nilim, and Laurent El Ghaoui. Robust control of Markov decision processes with uncertain transition matrices. Operations Research 53.5 (2005): 780-798.
>
> Cong Lu, Philip Ball, Jack Parker-Holder, Michael Osborne, and Stephen J. Roberts. Revisiting design choices in offline model-based reinforcement learning. International Conference on Learning Representations, 2022.
>
> Kurtland Chua, Roberto Calandra, Rowan McAllister, and Sergey Levine. Deep reinforcement learning in a handful of trials using probabilistic dynamics models. In Advances in Neural Information Processing Systems, 2018.
>
> Michael Janner, Justin Fu, Marvin Zhang, and Sergey Levine. When to trust your model: Model-based policy optimization. In Advances in Neural Information Processing Systems, 2019.
>
> Bradley Efron. The jackknife, the bootstrap and other resampling plans, volume 38. SIAM, 1982.

---

> ### Author Response · Authors · 2022-11-19
> **[2/2] Response to fgeG**
>
> ### "What makes CBOP perform well in situations where MOPO fails ..."
>
> Thank you for the excellent question. We note at least four differences between CBOP and MOPO that explains the difference in performance:
> 1. unlike MOPO, CBOP does not attempt to estimate an uncertainty measure over model or value predictions directly
> 2. CBOP trades off the uncertainty of model-based rollouts with that of value bootstraps, allowing it to make use of these two estimates more intelligently
> 3. CBOP uses model-based rollouts to estimate the target values during policy evaluation; MOPO, on the other hand, augments the replay buffer with synthetically generated rollout trajectories to train the policy.
> 4. The MVE, on which CBOP is based, is arguably less prone to the ensemble diversity issue than MOPO because of the difference stated in the last point.
>
>
> ### "..., the first term in equation (10) will start to fade."
>
> In fact, this is not technically correct. In the offline setting, the epistemic uncertainty of state-action pairs we do not observe in the training data (e.g., out-of-distribution points) will have high epistemic uncertainty. The ability to produce well-calibrated uncertainty estimates of Q-values and models, and use them for conservative value estimation, is one of the reasons why CBOP has performed so well empirically.
>
> To validate this hypothesis, we ran an additional experiment as suggested to validate the importance of the $A$ term in the total variance (e.g., the Q ensemble uncertainty). Please see Appendix D2 for the full details. Specifically, we used the CBOP agent trained on the random data set, and estimated the two uncertainty terms ($A$ and $B$) on batches from the expert data set. The idea is that a sample from the expert data set would be strictly out-of-distribution with respect to the random data since the random dataset consists entirely of state trajectories that do not deviate very far from the starting state (due to the failure of the agent), while the expert dataset consists of high-quality trajectories that indeed do. As a control, we also used the same agent to predict the random data (e.g., the training data) which is not OOD.
>
> The results are illustrated in Figure 8 in the Appendix. As we can see, the contribution of the $A$ term is much larger on OOD data than on the training data, suggesting that indeed it does not vanish on OOD data. Also, we see that the contribution of the $A$ term is greater (smaller) for a shorter (longer) roll-out horizon, which is intuitive since the longer horizon return should instead be dominated by model uncertainty. This justifies the use of both model and value uncertainty for adaptively tuning the weights in the MVE estimate, and we believe explains why CBOP works so well comparatively.
>
>
> ### "Why did you not extend the policy optimization to out-of-distribution data points ..."
>
> This is a good question. The main research question we tackled with CBOP was whether MVE-based approaches could work equally well in the offline setting. In principle, one approach to training on OOD data points is to augment the MVE target updating with imagined rollouts from the learned model. We believe that it will be crucial to learn to trust the generated data only when the ensemble estimates of the model agree with each other. This is an interesting extension of our framework that is left for future work.
>
> ### "Could you clarify the similarities/differences/benefits between your approach and theirs?"
>
> Thank you for sharing this work, which we were unaware of at the time of submission. Based on our understanding, both RAMBO-RL and CBOP take conservative approaches to learn the policy from offline data. However, based on our understanding one of the crucial differences is that RAMBO-RL achieves a conservative estimate of the value function by penalizing the model during training (e.g., training an adversarial model using a penalty on the value), whereas CBOP does this instead by penalizing the value function using the lower-confidence bound during policy optimization. Therefore, CBOP does not need to bias the model directly but instead uses well-calibrated Bayesian estimates of the model-return predictions to perform this penalization. The ability of CBOP to adapt the rollout horizon based on this uncertainty to our knowledge is not done in RAMBO-RL.

---

### Official Review · Reviewer_swCh · 2022-10-31

**Confidence:** 4
**Correctness:** 3
**Technical Novelty And Significance:** 2
**Empirical Novelty And Significance:** 2
**Recommendation:** 6

**Clarity, Quality, Novelty And Reproducibility:**

- The paper is well written, but as mentioned above, a clear algorithm box to describe the method would be helpful for better understanding.
- The novelty is somewhat limited, since the work largely extends a well known prior approach (MVE) from online RL to offline RL. The broad approach taken for this, namely conservative estimation, has also been explored in numerous prior works like MOReL, CQL, MOPO.
- That said, the specifics of getting the conservative estimates by tracking the full posterior and leveraging the LCB is new. However, at the writing level, this does not come across clearly.

**Strength And Weaknesses:**

- The paper can be interpreted as incorporating MVE to the offline RL setting. While this is incremental in nature, it nevertheless is a useful contribution and can be of interest to the offline RL community.
- Having a concrete algorithm box to describe the method can be useful to better understand the nuances of practical application.

**Summary Of The Paper:**

- Paper studies the offline RL setting, and specifically model-based approaches to offline RL.
- Paper hypothesizes that model-based value expansion (MVE) can help improve the quality of model-based offline RL algorithms. However, a naive application is found to be unsuccessful due to model bias which is particularly prominent in the offline RL setting.
- To overcome above challenge, the paper devices a conservative estimation procedure based on uncertainty quantification, similar in spirit to prior work like MOPO and MOReL.
- The paper empirically evaluates the proposed method (CBOP) on D4RL benchmark tasks and find significant improvements over prior methods.

**Summary Of The Review:**

Paper extends MVE to offline RL setting. While this is conceptually an incremental advancement, the proposed algorithm empirically displays improvements over prior work. Overall, I recommend a neutral to weak accept rating.

---

> ### Author Response · Authors · 2022-11-19
> **Response to Reviewer swCh**
>
> ### "The paper can be interpreted as incorporating MVE to the offline RL setting."
>
> We want to clarify that the incorporation of the MVE to the offline RL setting is not the only contribution of our paper. Typically, algorithms that work well in the online setting fail in the offline setting, because of the challenges associated with policy learning purely on offline data (e.g., overestimation, distribution shift), which is a nontrivial problem that MVE alone cannot address. CBOP novelly addresses this problem in more than one way. Firstly, it learns an adaptive weighting over $h$-step returns that is informed by the full posterior distributions over model and value estimates. This allows CBOP to perform longer (or arbitrary) horizon rollouts, in contrast to other model-based offline RL approaches where the horizon must be tightly restricted (Yu et al., 2020; Yu et al., 2021; Wang et al., 2021). Secondly, the more cautious use of the full distribution in the conservative estimate of the value functions is not present in MVE approaches, and this is the key to making CBOP successful in the offline regime as experimentally shown. We have updated our manuscript to better explain the differences between MVE and CBOP.
>
>
> ### "Having a concrete algorithm box ..."
>
> We appreciate the suggestion. We currently have an algorithm box in the appendix due to space constraints in the main text.
>
>
> ### "The novelty is somewhat limited, since the work largely extends a well known prior approach (MVE) from online RL to offline RL ..."
>
> As you correctly pointed out, prior offline RL work has demonstrated a clear necessity to be conservative for policy optimization to address the challenges of offline RL (distribution shift, value overestimation), and CBOP is motivated by this. Crucially, however, we want to reiterate that the main difference between CBOP and the existing literature is the tractable estimation of the full posterior distributions over model-based return predictions, and the conservative LCB that performs better than other value-based offline RL approaches, as demonstrated extensively in our experiments.
>
>
> ### "However, at the writing level, this does not come across clearly."
>
> Thank you for your comments about our paper. We have revised the introduction of the paper to make this point clearer.
>
>
> ### "MVE to offline RL setting. While this is conceptually an incremental advancement."
>
> We would like to reiterate the main advantages and contributions of CBOP:
> 1. apply a Bayesian approach to learn the model and target value uncertainty,
> 2. a novel adaptive weighting scheme inspired by MVE and the full distribution and
> 3. the use of the full distribution to derive a more conservative Bayesian value estimate based on the lower confidence bound, that outperforms existing conservative value-based offline RL.
>
>
> __References__
>
> Tianhe Yu, Garrett Thomas, Lantao Yu, Stefano Ermon, James Zou, Sergey Levine, Chelsea Fine, Tengyu Ma: MOPO: Model-based Offline Policy Optimization (2020).
>
> Tianhe Yu, Aviral Kumar, Rafael Rafailov, Aravind Rajeswaran, Sergey Levine, Chelsea Finn: COMBO: Conservative Offline Model-Based Policy Optimization (2021)
>
> Jianhao Wang, Wenzhe Li, Haozhe Jiang, Guangxiang Zhu, Siyuan Li, and Chongjie Zhang. Offline reinforcement learning with reverse model-based imagination. Advances in Neural Information Processing Systems, 34:29420–29432, 2021.

---

### Official Review · Reviewer_aFWb · 2022-11-02

**Confidence:** 4
**Correctness:** 4
**Technical Novelty And Significance:** 3
**Empirical Novelty And Significance:** 3
**Recommendation:** 8

**Clarity, Quality, Novelty And Reproducibility:**

The paper’s writing is fairly clear. I really like Figure 2, which provides a nice-looking and succinct summary of the algorithm.

The Bayesian formulation and resulting algorithm are novel, to my knowledge.

The full algorithm description and hyperparameters are included in the appendix. I believe it should be reproducible from the information provided.

**Strength And Weaknesses:**

Strengths:
* Jointly balancing the uncertainty in the model and Q function seems a good idea to me, and this paper presents a reasonably principled way to do that.
* Experimental results are good. CBOP closes the performance gap between model-based and model-free algorithms in the offline setting, and is demonstrably conservative w.r.t. the true returns.
* In the appendix, the authors provide evidence supporting their assumption of Gaussianity.

Weaknesses
* It is not obvious to me that the Bayesian formulation is necessary; the gains in performance could simply be from being conservative over the model and Q function jointly. A possible baseline algorithm to test this question would compute the target via a conservative quantile of all the values $\{\hat{R}_h^{k,m}\}$. This would be a more appropriate baseline than STEVE (which is not conservative because it was not designed for the offline setting), and if CBOP outperforms such an algorithm, the case for the Bayesian approach would be stronger.
* The comparison of CBOP-estimated values to actual returns is only performed for 3 datasets, all the same task. Including results for all (or at least more) tasks in the appendix would strengthen the reader’s confidence in the algorithm.
* The hyperparameter tuning sounds a bit dubious. The authors claim “The only hyperparameter that we have tuned is the LCB coefficient $\psi$“, but then they also state environment-dependent values for $M$ and $\eta$.

**Summary Of The Paper:**

The paper studies model-based RL in the offline setting. Offline RL typically requires regularization/penalty to avoid distributional shift and enable reasonably accurate value estimation. In contrast to previous model-based algorithms, which penalize only according to model uncertainty, this paper proposes a Bayesian-inspired conservative value estimation mechanism which incorporates both model and Q function uncertainty.

The authors describe how to approximate the posterior mean and variance of the Q values using ensemble methods. With these in hand, the algorithm can compute a conservative estimate of the Q value via the lower confidence bound. The proposed algorithm, CBOP, demonstrates strong performance on the D4RL benchmark and is (empirically) shown to provide conservative estimates of the actual returns.

**Summary Of The Review:**

I think the paper is a useful contribution overall. It presents a novel algorithm that appears effective and is well-backed with empirical support. However, the paper could be further strengthened with additional experiments, as detailed above.

---

> ### Author Response · Authors · 2022-11-19
> **Response to Reviewer aFWb**
>
> ### “It is not obvious to me that the Bayesian formulation is necessary.”
>
> Please note that unlike the quantile approach or other approaches investigated previously, the Bayesian approach in CBOP allows us to estimate the full posterior distribution, which can then be used to produce a variety of robust measures (e.g. quantiles, moments). We chose the lower confidence bound due to its intuitive interpretation, computational tractability, and theoretical justifications argued in related literature (see our discussion in the paper for details), but we clarify that this is not the only possible choice. We also ran an additional experiment to compare your quantile-based approach (see below).
>
> ### “Compute the target via a conservative quantile of all the values."
>
> This is an interesting idea! Conceptually, we argue that one shortcoming of taking a quantile approach is that it does not allow for a natural way to adapt the horizon/roll-out length, which is one of the benefits of CBOP.
>
> To verify the advantages of each in practice, we ran additional experiments comparing your suggested quantile approach with CBOP, and the findings are summarized in Appendix D.3 and Table 6. Note that due to limited computational resources, we have selected a subset of the tasks to evaluate the quantile baseline.
>
> To summarize, we found that it performed quite well in comparison to existing benchmarks, but that it failed to outperform the LCB estimate used in CBOP.  As argued above, one possible explanation for this is that the quantile estimator does not take into account the uncertainties associated with different-length horizons, while CBOP does, and this allows it to use the model more cautiously while avoiding value overestimation. We also found that CBOP produced much lower variability in actual performance than the quantile estimate.
>
> ### “... only performed for 3 datasets, all the same task.”
>
> Following your recommendation, we ran the experiment for all three tasks and three data sets {*m*, *mr*, *me*}. The results are summarized in Table 5 in the Appendix. In summary, we find that not only the mean predictions are negative but also the maximum values are, which affirms that CBOP indeed has learned conservative value functions. Also, the differences between the values computed by CBOP and the true discounted returns are much smaller in most of the scenarios. We hope this clarifies the benefits of bayesian conservatism proposed in CBOP.
>
>
> ### “The hyperparameter tuning sounds a bit dubious.”
>
> Following your suggestions, we revised the manuscript in Appendix C.2 to better explain the hyperparameter selection. In the reported experiments, we want to clarify that we did not tune $\eta$, but we kept $\eta = 1$ fixed throughout all benchmarks and configurations (the previous value of $50$ was a typo in our manuscript which we have corrected). For the parameter $M$, in the initial stage of testing the idea with the D4RL benchmark, we selected the medium configuration from all three environments as a baseline and tried $M=20$ which worked well on HalfCheetah and Walker2d environments without tuning (we wanted to keep it as low as possible for computational reasons). However, we found that we needed to have a larger value ensemble to get reasonable performance in Hopper. We chose $M=50$ following the recommendation of previous work (EDAC) and this worked well in our setting as well.
>
> The conservativeness parameter $\psi$ was tuned using a grid search in $[0.5, 2.0, 3.0, 5.0]$ using the online evaluation.  To justify the choice of these hyperparameters, we have additionally run some post hoc analysis based on the fitted Q-evaluation (FQE) algorithm, following the suggestion of Paine et al. (2020) to tune $\psi$ on offline data.  Table 4 summarizes the results, and it also shows that the rankings of the $\psi$ values we get through the online evaluation match very well with those we obtain from FQE, hinting that CBOP can be reliably tuned only using offline data.  An extensive discussion of further details surrounding this procedure has been added to Appendix C.2 (and Table 4) following your suggestion. We hope these added details clarify some of the confusion about how the hyperparameters were selected, and how the choices can be justified in the offline context.
>
> __References__
>
> Tom Le Paine, Cosmin Paduraru, Andrea Michi, Caglar Gulcehre, Konrad Żołna, Alexander Novikov, Ziyu Wang, Nando de Freitas: Hyperparameter Selection for Offline Reinforcement Learning (2020).

---

### Author Response · Authors · 2022-11-19
**Summary of Major Revisions**

We would like to thank the reviewers for their insightful and constructive comments. Below we provide a summary of all the major revisions made after reading the reviews provided.

*Novelty:*

For clarity in our discussion, we summarize the main contributions of CBOP over existing state-of-the-art offline model-based RL. Specifically, we contribute:
1. a Bayesian approach that expresses the uncertainty of model-based return predictions by leveraging Bayesian bootstrapped estimation of model and value functions
2. a novel adaptive weighting scheme of $h$-step returns inspired by MVE that is informed by the uncertainty estimation
3. the use of the full posterior distribution as a means of deriving a conservative value estimate based on the lower confidence bound
4. extensive empirical validation of all algorithmic design considerations, ablation studies, and comparison to the state-of-the-art demonstrating consistent performance improvement.

Below is a summary of our revisions and further empirical evaluations we performed following reviews.

*Writing and Clarity:*
1. We improved the introduction of the paper to better reflect the contributions of CBOP.
2. We revised the notation (e.g. $Q$ vs. $\hat{Q}$) in the probabilistic description of our algorithm (Section 3).
3. In the related work (Section 5), we added further details regarding the similarities and differences between CBOP and other works, and we added missing references that were suggested during the review.
5. In Appendix B.2, we clarified how the dynamics models were trained.
4. In Appendix C.2, we improved the discussion surrounding hyperparameter selection, particularly how $M$, $\eta$ and $\psi$ were selected.

*Empirical Studies:*
1. We compared CBOP to the baseline based on quantile estimation (DiMVE), and discuss the findings in Appendix D.3. We found that CBOP outperforms quantile estimation.
2. We added the missing comparison between the CBOP estimated values and the true returns (Table 5).
3. We performed further ablation to test the benefit of having both value and model uncertainty to estimate variance, and discuss findings in Appendix D.2. We found that the value uncertainty is indeed necessary when predicting on out-of-distribution data.

---

### Decision · Program_Chairs · 2023-01-20

**Decision:**

Accept: poster

**Justification For Why Not Higher Score:**

The proposed method is novel but its novelty is somewhat limited.

**Justification For Why Not Lower Score:**

The paper proposes a novel solution for an important problem with experiments to support the findings. It is well-written and it seems all reviewers like it.

**Metareview: Summary, Strengths And Weaknesses:**

The paper studies the important problem of offline policy optimization. The reviewers found the paper well-written, the proposed method novel, and the experimental results promising and satisfactory. I would suggest that the authors take the reviewers' comments into consideration when preparing the final version of the paper.

**Note From Pc:**

if the above contains the word "oral" or "spotlight" please see: "oral" presentation means -> notable-top-5% and "spotlight" means -> notable-top-25%. As stated in our emails, we are disassociating presentation type from AC recommendations